# Relationship between Preventive Health Behavior, Optimistic Bias, Hypochondria, and Mass Psychology in Relation to the Coronavirus Pandemic among Young Adults in Korea

**DOI:** 10.3390/ijerph19159620

**Published:** 2022-08-04

**Authors:** Dong-Suk Lee, Hyun-Ju Koo, Seung-Ok Choi, Ji-In Kim, Yeon Sook Kim

**Affiliations:** 1College of Nursing, Kangwon National University, Chuncheon 24341, Korea; 2Hallym University Chuncheon Sacred Heart Hospital, Chuncheon 24253, Korea; 3Kangwon National University Hospital, Chuncheon 24289, Korea; 4Department of Nursing, California State University San Bernardino, San Bernardino, CA 92407, USA

**Keywords:** optimistic bias, hypochondriasis, mass psychology, preventive behavior, COVID-19, influenza

## Abstract

The great challenge to global public health caused by the coronavirus pandemic has lasted for two years in Korea. However, Korean young adults seem less compliant with preventive health behaviors than older adults. This study aims to explore the relationship between risk perception variables of optimistic bias, hypochondriasis, and mass psychology, and preventive health behavior in relation to the coronavirus pandemic through a cross-sectional online survey. The participants are 91 Korean young adults aged 19–30. The results show that mass psychology has a positive relationship with preventive health behavior, whereas optimistic bias and hypochondriasis do not. In detail, people with high or middle levels of mass psychology displayed higher preventive health behavior compared with those who had low levels of mass psychology, and the highest compliance was for wearing a mask, followed by COVID-19 vaccination, whereas the lowest compliance was for influenza vaccination. These findings could be explained by the Korean culture of strong collectivism and the characteristics of COVID-19, which evoked extreme fear globally. The results of this study can be useful for policy establishment in the ongoing prevention of COVID-19 and suggest that mass psychology should be used effectively in planning preventive communication campaigns.

## 1. Introduction

COVID-19 and influenza (flu), the illnesses causing the pandemic, are both contagious respiratory illnesses. The coronavirus pandemic has been an especially great challenge for global public health in the last two years. The World Health Organization (WHO) estimates that 1 billion people worldwide are infected with influenza annually. As of 3 August 2022, the WHO reported 577,018,226 cases and 6,401,046 deaths globally from COVID-19 [1]. In South Korea, the first confirmed case appeared in February 2020, after the first cases appeared in China in late 2019, and the Korea Disease Control and Prevention Agency (KDCA) reported 20,052,305 cases and 25,110 deaths as of 3 August 2022 [2].

To remain safe during this public health crisis, individuals must take preventive measures. Accordingly, mass media report news related to COVID-19 to guide the public and promote preventive measures, including vaccination campaigns, every day. However, young adults seem less compliant with preventive behaviors than older adults, as COVID-19 is known to be more fatal to older adults. Based on a precedent study [3], being younger was associated with a lower chance of adopting the preventive rules during the H1N1 influenza pandemic. Young adulthood is an important period for establishing health behavior patterns in life.

Health behavior theories suggest that perceived risk is a key determinant of engagement in preventive behaviors. Perceived severity, a kind of risk perception, was a factor related to preventive behaviors during the COVID-19 pandemic [4]. When people encounter health information, they mostly either experience general fear or concerns about relevant health crises in their imagination or think they may have a health crisis, which is called personalized risk perception. Risk perception is an individual’s level of perceived risk in a message and is one of the main concepts supporting health behavior change in a person who receives a health promotion message [5]. Optimistic bias is the lack of the idea that a health crisis would happen to oneself, which means that there is no personalizing risk perception [6,7]. This bias occurs in the process of perceiving risk through a social comparison process, rather than being a simple belief [8]. Optimistic bias may decrease anxiety and provide psychological benefits for emotional well-being [8,9,10], but it has a negative relationship with preventive health behavior. A recent study on COVID-19-related health behavior showed that people with a high level of optimistic bias had a low-risk perception of contracting COVID-19, which negatively influenced their responses to health crises, information-seeking intentions, and behavior [9]. In addition, optimistic bias has a negative relationship with the adoption of protective behavior [10].

In contrast to optimistic bias, some people are excessively worried about health, which may cause a cognitive bias to develop health anxiety, which is called hypochondria or hypochondriasis [11]. Hypochondria is characterized by extensive worries about health [12] and lies on a continuous spectrum from clinical hypochondriasis to simple health anxiety [13]. The Diagnostic and Statistical Manual of Mental Disorders-V (DSM-V) is applied to hypochondria, which is classified as a somatic symptom disorder or an illness anxiety disorder [14]. However, hypochondriasis is viewed as a cognitive and perceptual impairment in psychology [15]. Lee [16] suggests that the illness attitude scale (IAS) is a suitable instrument for measuring hypochondriasis and includes four subscales: disease phobia and beliefs; impaired adaptation due to disease; safety pursuit behavior; and thanatophobia. Safety pursuit behavior, which is a factor in hypochondriasis, is related to preventive health behavior.

Another interesting factor influencing preventive health behavior is mass psychology, which is one’s perception based on the number of people exposed to the same risk as an individual. In a study of risk perception, Yamaguchi [17] reported that an individual’s perceived risk decreases as the number of people exposed to the same risk increases. This means that people feel secure when many people experience the same risk, which is called the group diffusion effect [18,19]. According to Rubin et al., who studied behavioral change during the swine flu outbreak [20], this is called public perception. In a recent study conducted during the COVID-19 pandemic, subjective norms related to parents was a significant factor associated with preventive behaviors among Korean young adults [21]. These studies indicate that one’s perception of the norm of the group they belong to, that is, mass psychology, is related to preventive health behavior.

Within the context of the COVID-19 pandemic, it is crucial to understand psychological factors, such as risk perceptions of preventive behaviors, to manage infection. Therefore, this study aims to explore the relationship between risk perception variables of optimistic bias, hypochondriasis, and mass psychology, and preventive health behavior in relation to the coronavirus pandemic in young adults.

## 2. Materials and Methods

### 2.1. Study Design

This cross-sectional descriptive study via an online survey examined the relationship between optimistic bias, hypochondriasis, mass psychology, and preventive health behavior for COVID-19 or influenza in young adults.

### 2.2. Participants and Enrollment

The study participants were recruited through an online posting on K university’s website from the 31 May to the 6 June 2022. Inclusion criteria were as follows: young adults aged 19–30 years who agreed to participate in the study after understanding the study purpose, and undergraduate or graduate students in K university. Exclusion criteria were those who were hospitalized or immobile with severe diseases. The sample size was calculated using the G power 3.1.9.2 program. With a level of significance (α) of 0.05, a medium effect size of 0.3, and a power of 0.80, the required sample size was 82. A total of 91 participants participated in the study.

### 2.3. Measurement

Four instruments were used in this study; optimistic bias (4 questions); hypochondriasis (27 questions); mass psychology (3 questions); and preventive health behaviors (6 questions). Questions on general characteristics of the study subject included gender, age, experience of COVID-19, experience of illness, history of hospitalization, religion, present illness, and cell phone number.

#### 2.3.1. Optimistic Bias

Optimistic bias can be measured using two risk determinants: comparative risk and absolute risk. Comparative risk was assessed using the question ’compared with the average person of your age and sex, are you more, equally, or less likely to experience X?’. Absolute risk was assessed using two questions: ‘what is the possibility of a health crisis X you will have? (X1)’ and ‘what is the possibility of the health crisis X in those who are the same age and sex as you? (X2)’. If the difference between X2 and X1 (X2–X1) is positive, it is interpreted as an optimistic bias. This means that the person has an optimistic bias when giving a more favorable rating than other people.

Clarke et al. [22] support the use of absolute risk to reduce comparative consciousness and noticeable bias as a more conservative measure than using comparative risk. Accordingly, this study used absolute risk as the determinant of optimistic bias. In this study, optimistic bias for susceptibility was measured by asking a set of two questions: ‘what is the possibility of health crisis X (COVID-19 or influenza) you will have?’ and ‘what is the average possibility of health crisis X in those of the same age and sex as you?’ Optimistic bias for severity was measured by asking a set of two questions: ‘what is the chance you die of the health crisis X?’ and ‘what is the average chance to die of health crisis X in those of the same age and sex as you?’. The score for each question ranged from 1 point (very low) to 4 points (very high). It was interpreted that optimistic bias existed when the difference between the average score of other people and the average score of the subject (X2 − X1) was positive. In addition, optimistic bias was considered greater when the sum of the two differences was larger. In this study, Cronbach’s α was 0.72.

#### 2.3.2. Hypochondriasis

The Illness Attitude Scale (IAS) was developed by Kellner [23] to measure hypochondriasis. This scale comprises a total of 27 question items with 9 categories, including worry about illness, concerns about pain, health habits, hypochondriacal beliefs, thanatophobia, disease phobia, bodily preoccupations, treatment experience, and effects of symptoms. Lee [24] translated and validated the scale to measure hypochondriac fear, beliefs, and attitudes. Lee [16] conducted a factor analysis of the scale and categorized it into 4 subscales: disease phobia and beliefs; impaired adaptation due to disease; safety pursuit behavior; and thanatophobia. Each subscale score ranged from 0 (strongly disagree) to 4 (strongly agree), with a higher score indicating a greater tendency for hypochondriasis. Lee [13] suggests that IAS is a suitable instrument for measuring hypochondriasis. Cronbach’s α was 0.86 in the study by Lee [16] and 0.91 in this study.

#### 2.3.3. Mass Psychology

Mass psychology was assessed by measuring one’s perception of other people’s handwashing, mask-wearing, and vaccinating. Each item score ranged from 0 (strongly disagree) to 4 (strongly agree), with a higher score indicating greater mass psychology, that is, one perceives a larger group of people performing preventive health behavior. Cronbach’s α was 0.75 in a study by Lee [19], and 0.78 in this study.

#### 2.3.4. Preventive Health Behavior

Preventive health behavior was assessed by measuring compliance with 6 preventive health behaviors related to influenza and COVID-19, including handwashing, wearing a mask, social distancing, COVID-19 vaccination, influenza vaccination, and proper fluid intake, including humidification. This instrument was made based on the preventive measures for COVID-19 suggested by the WHO, which include 6 items. All items were evaluated for content validity by 3 nursing professors on a scale of 1 to 4 points. The content validity index (CVI) for all 6 items was 3 points or higher. Exploratory factor analysis conducted for construct validity revealed that all 6 items had communality of 0.5 or higher, with a KMO (Kaiser–Meyer–Olkin) of 0.704, Bartlett χ^2^ of 124.072, df of 15, and *p* of 0.000, which showed the appropriateness of the instrument. Each item score ranged from 0 (strongly disagree) to 4 (strongly agree), with higher scores indicating more preventive health behaviors. In this study, Cronbach’s α was 0.74.

### 2.4. Data Collection and Ethical Consideration

This study was approved by the institutional review board of K University before data were collected to protect the study participants, who were recruited through the posting on the K University online site and were asked to complete the survey online. The study participants voluntarily signed the informed consent form online after reviewing the instructions on the consent for the study subject and then proceeded with the survey. They were informed that the completion of the survey would take 10 min and were given the lead researcher’s contact information (cell phone and email address) for further questions in the instructions. Participants’ responses, including personal information, were collected in the secured folder of the lead researcher’s computer drive to protect participants’ sensitive information. Each participant in the study received a KRW 5000.00 (approximately USD 5.00) Starbucks gift card via their cell phone as gratitude for their contribution, and then all personal information of the participants was removed.

### 2.5. Statistical Analysis

Data were analyzed using the IBM SPSS software (version 26.0; IBM Corp., Armonk, NY, USA). Descriptive statistics were used to analyze the general characteristics of the study participants. Normality was tested using Kolmogorov–Smirnov and Shapiro–Wilk.

The differences in optimistic bias, hypochondriasis, mass psychology, and preventive health behavior based on participants’ characteristics (gender, age, experience of COVID-19, experience of illness, history of hospitalization, and present illness) were analyzed using Mann–Whitney, a non-parametric test for a non-normal distribution of variables: optimistic bias, hypochondriasis, mass psychology, and preventive health behavior. The differences in preventive health behavior among the three groups (lower than 25%, 25–75%, and higher than 75%), based on the level of optimistic bias, hypochondriasis, and mass psychology, were analyzed using the Kruskal–Wallis test, a non-parametric test. The correlation between optimistic bias, hypochondriasis, mass psychology, and preventive health behavior was analyzed using Spearman’s rho, a non-parametric test.

## 3. Results

### 3.1. General Characteristics

The study participants’ characteristics were age, gender, experience of COVID-19, experience of illness (including COVID-19), present illness, and history of hospital administration. There were 91 participants, with 38 males (41.8%) and 53 females (58.2%), and the average age was 25.40 years, with 27 participants younger than 25 years (30%) and 63 participants older than 26 years (70%). All participants were divided into two groups, younger than 25 years and older than 26 years, since the mean value (25.40) is located between 25 and 26 years. COVID-19 was contracted by 54.9% of the participants, illness, including COVID-19, was experienced by 73.6% of the participants, prior hospitalization was experienced by 57.1% of the participants, and the present illness was experienced by 11% of the participants. The variables of preventive health behavior, optimistic bias, hypochondria, and mass psychology showed a non-normal distribution (Table 1).

Table 2 shows compliance with six preventive health behaviors in young adults. The highest compliance was observed in wearing a mask, followed by the COVID-19 vaccination, whereas the lowest compliance was with influenza vaccination.

### 3.2. Difference between Optimistic Bias, Hypochondriasis, and Mass Psychology Associated with Participants’ Characteristics

The Mann–Whitney test showed differences in the main variables (optimistic bias, hypochondriasis, mass psychology, and preventive health behavior) based on the general characteristics of the participants. People without present or previous illness showed significantly higher optimistic bias scores than those with illness. In addition, individuals with COVID-19 or other illnesses showed significantly higher hypochondriasis scores than those without illnesses. Those aged 26 years or older had significantly higher mass psychology scores than those aged 25 years or younger. However, preventive health behaviors were not associated with the general characteristics of the participants (Table 3).

### 3.3. Relationship between Preventive Health Behavior, Optimistic Bias, Hypochondria, and Mass Psychology

Correlations between the main variables (preventive health behavior, optimistic bias, hypochondria, and mass psychology) show that there was only a significant relationship between preventive health behavior and mass psychology (Table 4). This means that participants with a high level of mass psychology showed a high level of preventive health behavior (Figure 1).

### 3.4. Preventive Health Behavior by Optimistic Bias, Hypochondria, and Mass Psychology

Differences in preventive health behavior based on optimistic bias, hypochondria, and mass psychology were analyzed using the Kruskal–Wallis test. Preventive health behaviors did not show differences based on the level of optimistic bias or hypochondria. However, the results differed according to the level of mass psychology. Post hoc tests revealed that the groups with high (top 25%) and middle (25–75%) levels of mass psychology showed a higher score for preventive health behavior than the group with a low level (bottom 25%) of mass psychology (Table 5).

## 4. Discussion

This study aims to explore the relationship between preventive behaviors, optimistic bias, hypochondria, and mass psychology within the context of the coronavirus pandemic among Korean young adults. Risk perception in Korean young adults showed that individuals without past illness, including COVID-19, had a lower optimistic bias and higher hypochondriasis than those without it. These findings are similar to the study results on optimistic bias about cervical cancer in Korean female college students, stating that optimistic bias diminished as the individual’s involvement in cervical cancer was enhanced [25]. Trobst et al. [26] reported that related experience is positively associated with risk perception. Weinstein [6] stated that people who had not experienced illness or health problems were likely to be optimistic about perceiving health and safety risks, whereas those with past illness or health problems tended to be hypochondriac, characterized by worrying about general health.

Interestingly, our study identified that mass psychology was related to preventive health behavior regarding influenza and COVID-19 but not optimistic bias or hypochondriasis. Mass psychology showed a positive relationship with preventive health behavior, and people with high or middle levels of mass psychology had higher preventive health behavior than those with low levels of mass psychology. These findings are similar to the study result by Park et al., who reported that subjective norms related to parents were a significant factor influencing Korean young adults’ preventive behavior, but perceived susceptibility and severity were not [21].

This may be interpreted in two ways. The first explanation is Korean culture. Korea has a higher tendency for collectivism [27]. Koreans’ strong collectivism helps to solve problems and does not allow an individual’s behavior to cause a conflict or crisis in others or a group of people [28], and an individual is likely to behave like most people in a group. Accordingly, it is interpreted that a person tends to follow other people’s behaviors when the individual recognizes that many people engage in preventive health behaviors around them.

Park and Kim [27] identified Koreans’ cultural self-orientation using a meta-analysis of 41 national studies and reported that Koreans were more likely to be collectivist than individualist as they aged. Our study showed a higher score for mass psychology, especially in those aged 26 years or older compared to those at aged 25 years or younger, which was in accordance with previous studies. In addition, unlike in Western countries, mask-wearing was mostly complied with among all six preventive health behaviors, which can also be explained due to this aspect of Korean culture.

The second explanation is the characteristics of COVID-19. It is probably one of the reasons why optimistic bias or hypochondriasis were not associated with preventive health behavior for influenza and COVID-19 in Korean young adults. The COVID-19 pandemic has evoked extreme fear globally, and societal prevention guidelines would have more influence on preventive health behavior than the person’s risk perception. Although previous studies supported the relationship between risk perception variables, such as optimistic bias or hypochondriasis, and preventive health behavior [9,10,29,30,31], there are other studies showing no relationship [32,33]. Therefore, further studies on risk perception and preventive health behavior need to be conducted, and more studies are necessary to identify the factors that predict preventive health behavior for influenza or COVID-19. The limitations of this study include the failure to conduct a regression analysis to identify factors predicting preventive health behavior due to the non-normal distribution of major variables, despite the sample size of 91 participants.

## 5. Conclusions

This study concludes that preventive health behaviors for COVID-19 and influenza are associated with mass psychology, that is, the perceived group size in Korean young adults. This may result from Koreans’ strong collectivism and the global phobia that COVID-19 caused. The findings of the study provide useful implications for establishing policies to prevent COVID-19, which are still in progress, and suggest that mass psychology should be effectively applied in planning health communication campaigns for public safety.

## Figures and Tables

**Figure 1 ijerph-19-09620-f001:**
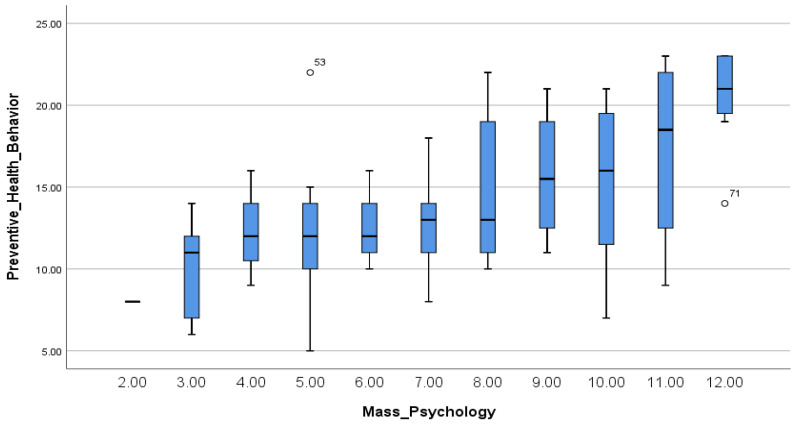
Spearman’s correlations between preventive health behavior and mass psychology.

**Table 1 ijerph-19-09620-t001:** Characteristics of participants (*N* = 91).

Variables	Mean ± SD	Category	*n* (%)	Normality Test †	
				Kolmogorov–Smirnov (*p*)	Shapiro–Wilk (*p*)
Age (year)	25.40 ± 2.79	≤25	27 (30.0)		
		≥26	63 (70.0)		
Gender		Male	38 (41.8)		
		Female	53 (58.2)		
Experience of COVID-19		Yes	50 (54.9)		
		No	41 (45.1)		
Experience of illness (including COVID-19)		Yes	67 (73.6)		
		No	24 (26.4)		
Present illness		Yes	10 (11.0)		
		No	81 (89.0)		
History of hospital administration		Yes	52 (57.1)		
		No	39 (42.9)		
Preventive health behavior †	14.15 ± 4.42	Lower than 25% (≤11)	28 (30.8)	0.130 (0.001)	0.962 (0.010)
		25–75% (12–16)	36 (39.6)		
		Higher than 75% (≥17)	27 (29.7)		
Optimistic bias †	0.34 ± 1.12	Negative (−)	9 (9.9)	0.290 (0.000)	0.830 (0.000)
		Neutral (0)	52 (57.1)		
		Positive (+)	30 (33.0)		
Hypochondria †	30.73 ± 13.54	Lower than 25% (≤21)	23 (25.3)	0.108 (0.011)	0.948 (0.001)
		25–75% (22–37)	45 (79.5)		
		Higher than 75% (≥38)	23 (25.3)		
Mass psychology †	7.58 ± 2.49	Lower than 25% (≤6)	33 (36.3)	0.100 (0.026)	0.968 (0.023)
		25–75% (7–9)	35 (38.5)		
		Higher than 75% (≥10)	23 (25.3)		

† Four main variables do not have normal distributions because *p* values are less than 0.05.

**Table 2 ijerph-19-09620-t002:** Descriptive statistics of preventive health behavior (*N* = 91).

PHB	Mean	SD	Median	Mode
PHB1 (hand washing)	1.93	1.02	2	3
PHB2 (wearing mask)	3.10	0.97	3	4
PHB3 (social distancing)	2.16	1.13	2	2
PHB4 (hydration)	1.79	1.25	2	1
PHB5 (influenza vaccination)	1.65	1.44	1	0
PHB6 (COVID-19 vaccination)	2.53	0.77	3	3

PHB = preventive health behavior; SD = standard deviation.

**Table 3 ijerph-19-09620-t003:** Difference of main variables by participant characteristics (*N* = 91).

Variables	Category	Preventive Health Behavior		Optimistic Bias		Hypochondria		Mass Psychology	
		M ± SD	Z (*p*) †	M ± SD	Z (*p*) †	M ± SD	Z (*p*) †	M ± SD	Z (*p*) †
Age (year)	≤25	14.04 ± 4.51	−0.049 (0.961)	0.40 ± 1.18	−0.504 (0.614)	33.25 ± 14.87	−0.877 (0.381)	6.63 ± 2.59	−2.514 (0.012)
	≥26	14.11 ± 4.38		0.31 ± 1.10		29.43 ± 12.89		7.97 ± 2.38	
Gender	Male	14.26 ± 4.98	−0.065 (0.948)	0.45 ± 1.20	−0.036 (0.971)	30.21 ± 12.77	−0.125 (0.901)	7.79 ± 2.51	−0.596 (0.551)
	Female	14.08 ± 4.01		0.26 ± 1.06		31.09 ± 14.18		7.43 ± 2.50	
Experience of COVID-19	Yes	13.62 ± 4.09	−1.161 (0.246)	0.22 ± 0.91	−1.597 (0.110)	33.50 ± 13.88	−2.567 (0.010)	7.18 ± 2.41	−1.326 (0.185)
	No	14.80 ± 4.77		0.49 ± 1.33		27.34 ± 12.46		8.07 ± 2.53	
Experience of Illness (including COVID-19)	Yes	13.84 ± 4.19	−1.075 (0.282)	0.16 ± 1.07	−2.592 (0.010)	33.31 ± 13.97	−3.259 (0.001)	7.28 ± 2.42	−1.737 (0.082)
	No	15.04 ± 4.98		0.83 ± 1.13		23.50 ± 9.17		8.42 ± 2.57	
Present illness	Yes	14.40 ± 5.30	−0.006 (0.995)	−0.60 ± 1.71	−2.109 (0.035)	36.20 ± 19.40	−0.086 (0.388)	7.70 ± 2.67	−0.064 (0.949)
	No	14.12 ± 4.34		0.46 ± 0.98		30.05 ± 12.64		7.57 ± 2.49	
History of hospital administration	Yes	14.09 ± 3.70	−0.402 (0.687)	0.23 ± 1.26	−1.015 (0.310)	30.33 ± 13.38	−0.490 (0.624)	7.73 ± 2.02	−0.590 (0.555)
	No	14.23 ± 5.28		0.49 ± 0.88		31.26 ± 13.91		7.38 ± 3.03	

† Mann–Whitney test.

**Table 4 ijerph-19-09620-t004:** Correlations of main variables (*N* = 91).

	Preventive Health Behavior	Optimistic Bias	Hypochondria	Mass Psychology
Preventive health behavior	1	−0.072 (0.495)	0.188 (0.075)	0.509 (0.000) †
Optimistic bias		1	−0.111 (0.293)	0.02 (0.868)
Hypochondria			1	−0.017 (0.875)
Mass psychology				1

† Values are Spearman’s rank correlation coefficient with *p* value.

**Table 5 ijerph-19-09620-t005:** Preventive health behavior by optimistic bias, hypochondria, mass psychology.

			Preventive Health Behavior			
	Group	*n* (%)	Mean Rank ± SE	χ^2^ †	*p*	Post Hoc
Optimistic bias	Negative (−)	9 (9.9)	14.89 ± 1.41	0.663	0.718	
	Neutral (0)	52 (57.1)	14.27 ± 0.59			
	Positive (+)	30 (33.0)	13.73 ± 0.87			
Hypochondria	Lower than 25%	23 (25.3)	12.87 ± 0.94	4.439	0.109	
	25–75%	45 (79.5)	14.22 ± 0.62			
	Higher than 75%	23 (25.3)	15.30 ± 0.98			
Mass psychology	Lower than 25% ^a^	33 (36.3)	11.82 ± 0.58	18.348	0.000	a < b, a < c, b = c
	25–75% ^b^	35 (38.5)	14.34 ± 0.64			
	Higher than 75% ^c^	23 (25.3)	17.22 ± 1.00			

† Kruskal–Wallis test; a, b, c = Bonferroni’s multiple comparison for Post-hoc analysis.

## Data Availability

The datasets used and/or analyzed in this study are available from the corresponding author on reasonable request.

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
