# Peer review of "Relationship between Preventive Health Behavior, Optimistic Bias, Hypochondria, and Mass Psychology in Relation to the Coronavirus Pandemic among Young Adults in Korea"

_ijerph, 2022, doi:10.3390/ijerph19159620_

Round 1

Reviewer 1 Report

It is a topic and a very interesting, current and theoretically and socially relevant work. However, the article needs to be improved in many ways before it can be recommended for publication:

1) The background must be updated with the most recent references and empirical evidence (2022) and remove old references. In addition, the background must conclude in the research question; specified in the objective and materialized in the hypotheses or expected forecasts.

2) The presentation of the methodology should be significantly improved. --- The participants must be clear, including in a nested way the number x gender x other characteristics; In addition, the inclusion and exclusion criteria must be made explicit; the representativeness of the sample, etc.

3) The description of the instruments must be complete, providing a table with them and their factorial weights, for example. In addition, with the study data itself, coefficients on composite reliability (not only for internal consistency or Cronbach's alpha), MacDonald's omega, the average variance extracted (AVE or convergent validity) must be provided; discriminant validity (square root of the convergent and that its coefficients are greater than the intercorrelations between the factors). Evidence of construct validity should be provided with the data itself, and if possible, confirmatory factorial analysis.

4) A section should be introduced prior to the results, on data analysis and software used, but with more explanatory arguments, greater detail, why these analyzes are used and not others, and what they contribute. In this sense, presenting data that "go together", pure correlations, does not allow any statement to be made about the causal relationship between the measures. It would be desirable for them to carry out a causal analysis: I suggest two possible alternatives; One could be through the Hayes PROCESS macro, which is inserted in the Regression Analysis module and uses bootstrapping, and is open access and very easy to implement; this would make it possible to provide mediation, moderation or conditional analyses. Another alternative, through AMOS (or another program), through SEM (structural equations); that would allow establishing a model of causal relationships that would be much more interesting and generalizable than what they provide in the study.

5) The presentation of results should improve notably. The steps indicated and explained before in the Data Analysis section should be followed. Of course, table 1 (and several of the following) is a simple list; so they don't do much for explanatory purposes. You should consider presenting complete nested tables, including not only the mean or median scores, statistical significances, effect sizes (practical value of relationships); and above all, to completely reorganize it to answer the research question that should allow causal analyzes (SEM type, PROCESS type?).

6) The Discussion and conclusions must be reorganized and rewritten. It must answer the research question or problem of the study; it must indicate if the objective was achieved; whether or not the hypotheses are confirmed. An interpretation must be made in line with recent references and studies (2022); the limitations and their way of resolving them must be assessed; applications for practice; and the added value of the study or conclusions.

7) The abstract must be rewritten according to these changes, clearly indicating the sample (N), what was done, the results and the added value. They must assess whether to also adjust the title to the new changes in the article.

8) Review all the formal and editorial aspects, articulating the article much more in its various parts. For example, update to the last date you review the article (or the latest possible before its publication) the reference 1 on the statistics used

9) All changes in the new version must be marked in color, to facilitate their verification. In addition, to argue one by one in the magazine's forum for the reviewer; so that what has been done can be properly understood and valued.

Author Response

Reviewer’s Comments and Suggestions for Authors

It is a topic and a very interesting, current and theoretically and socially relevant work. However, the article needs to be improved in many ways before it can be recommended for publication:

1) The background must be updated with the most recent references and empirical evidence (2022) and remove old references. In addition, the background must conclude in the research question; specified in the objective and materialized in the hypotheses or expected forecasts.

Response: We, researchers, truly appreciate your comments. According to your feedback, references are updated. However, some old references are used to define or explain terminologies, which are very crucial studies and remained in the revised manuscript. Also, they are classical references, mainly original studies of development of instruments, which need to be cited in the manuscript. Also, this study aimed to explore the relationship between variables, hypothesis was not created. Yet, we wrote the study objective more clearly as we respect your advice.

2) The presentation of the methodology should be significantly improved. --- The participants must be clear, including in a nested way the number x gender x other characteristics; In addition, the inclusion and exclusion criteria must be made explicit; the representativeness of the sample, etc.

Response: Inclusion and exclusion criteria were added in the section of materials and methods. Detailed description of participants’ characteristics was also written in results.

3) The description of the instruments must be complete, providing a table with them and their factorial weights, for example. In addition, with the study data itself, coefficients on composite reliability (not only for internal consistency or Cronbach's alpha), MacDonald's omega, the average variance extracted (AVE or convergent validity) must be provided; discriminant validity (square root of the convergent and that its coefficients are greater than the intercorrelations between the factors). Evidence of construct validity should be provided with the data itself, and if possible, confirmatory factorial analysis.

Response: Thank you for your comments. There are four tools used in the study:  assessment of Optimistic bias, Hypochondria, Mass psychology, and preventive health behavior. Three out of four (Optimistic bias, Hypochondria, & Mass psychology) were not developed in this study. They were developed before and have been used in many studies. Therefore, Cronbach's alpha for each tool was written.  For the tool to measure preventive health behavior, researchers made 6 question items after reviewing the COVID-19 related preventive measures of WHO. Content validity and construct validity were tested and accepted to use. Exploratory Factor analysis for Preventive Health Behavior was conducted, and two factors were shown. However, these were COVID-19 related preventive health behaviors, which were not labeled. Researchers thought this content was to be explained and added ‘This instrument was made based on the preventive measures on COVID-19 suggested by the WHO, which included 6 items. All items were evaluated for content validity by 3 nursing professors on a scale of 1 to 4 points. Content validity index (CVI) of all 6 items was 3 points or higher. Exploratory factor analysis conducted for construct validity re-vealed that all 6 items had communality of 0.5 or higher, with KMO (Kaiser-Meyer-Olkin) of .704, Bartlett χ2 of 124.072, df of 15, and p of .000, which showed the appropriateness of the instrument.’ in the revised manuscript. Although we did not add the table of Exploratory Factor analysis for Preventive Health Behavior in the manuscript, it is attached here for your information.

4) A section should be introduced prior to the results, on data analysis and software used, but with more explanatory arguments, greater detail, why these analyzes are used and not others, and what they contribute. In this sense, presenting data that "go together", pure correlations, does not allow any statement to be made about the causal relationship between the measures. It would be desirable for them to carry out a causal analysis: I suggest two possible alternatives; One could be through the Hayes PROCESS macro, which is inserted in the Regression Analysis module and uses bootstrapping and is open access and very easy to implement; this would make it possible to provide mediation, moderation, or conditional analyses. Another alternative, through AMOS (or another program), through SEM (structural equations); that would allow establishing a model of causal relationships that would be much more interesting and generalizable than what they provide in the study.

Response: Researchers aimed to identify causal relationship by using regression analysis, but found out that planned to conduct this study, four main variables (preventive health behavior, optimistic bias, hypochondria, mass psychology) didn’t have normal distributions based on Kolmogorov-Smirnov and Shapiro-Wilk tests. Accordingly, parametric test was not applicable and non-parametric test was used. Mann-Whitney test was used instead of T-test, Kruskal-Wallis test instead of ANOVA, and Spearman’s rho test instead of Pearson correlation. Regress analysis and SEM you suggested can be used when data are normally distributed as one of assumptions, which was not applicable to the data of this study. However, we attached the result of various regression analyses for your information. If you can provide rationales that can override this fundamental assumption for regression analysis, we are willing to add the results from regression analyses. Since the study could not identify causal relationship due to the data characteristics, the title of the manuscript was changed to ‘Relationship of preventive health behavior, optimistic bias, hypochondria, and mass psychology among young adults in Korea’. We also restricted expression that implicated causal relationship in the manuscript. Please refer to the Data analysis section and limitation of study in discussion that addressed this issue.

5) The presentation of results should improve notably. The steps indicated and explained before in the Data Analysis section should be followed. Of course, table 1 (and several of the following) is a simple list; so they don't do much for explanatory purposes. You should consider presenting complete nested tables, including not only the mean or median scores, statistical significances, effect sizes (practical value of relationships); and above all, to completely reorganize it to answer the research question that should allow causal analyzes (SEM type, PROCESS type?).

Response: There is no change in the analysis method for the above reason in (4). Please refer to the response to (4).

6) The Discussion and conclusions must be reorganized and rewritten. It must answer the research question or problem of the study; it must indicate if the objective was achieved; whether or not the hypotheses are confirmed. An interpretation must be made in line with recent references and studies (2022); the limitations and their way of resolving them must be assessed; applications for practice; and the added value of the study or conclusions.

Response: References were updated and the entire manuscript was thoroughly reviewed and revised.

7) The abstract must be rewritten according to these changes, clearly indicating the sample (N), what was done, the results and the added value. They must assess whether to also adjust the title to the new changes in the article.

Response: The abstract was reviewed, and the title was changed to be consistent with the findings of the study.

8) Review all the formal and editorial aspects, articulating the article much more in its various parts. For example, update to the last date you review the article (or the latest possible before its publication) the reference 1 on the statistics used.

Response: Thank you for your feedback. The last date of reviewing the article and the access date will be updated right before its publication.

9) All changes in the new version must be marked in color, to facilitate their verification. In addition, to argue one by one in the magazine's forum for the reviewer; so that what has been done can be properly understood and valued.

Response: The revised areas of the manuscript were written in red with track changes for you to easily review. Also, response was written under each comment of reviewers as seen.

We appreciate all you support indeed.

Reviewer 2 Report

This paper investigates the important topic related to relationship between 14 risk perception variables of optimistic bias, hypochondriasis, and mass psychology and preventive 15 health behavior to the pandemic coronavirus through a cross-sectional survey.

·      The objective of the study is clear 

·      Abstract appropriately summarizes the study

·      Results are clearly explained with tables

·      Discussion part is meaningful

Revision Comments

1.     When was this study conducted? How long? Time period/ year/ duration of the study?

2.     In the abstract, authors have mentioned age 19-30 whereas in method section it says 18-30. Please correct it 

3.     Is there a rationale of selecting the particular age group?

4.     Mention about how  participants confidently was maintained?

5.     In methods section, there is a mention of self-reported questionnaire. Is this adopted from somewhere? Or is it self-made? You can add questionnaire as appendix. 

Author Response

Reviewer’s Comments and Suggestions for Authors

This paper investigates the important topic related to relationship between 14 risk perception variables of optimistic bias, hypochondriasis, and mass psychology and preventive 15 health behavior to the pandemic coronavirus through a cross-sectional survey.

  • The objective of the study is clear
  • Abstract appropriately summarizes the study
  • Results are clearly explained with tables
  • Discussion part is meaningful

1) When was this study conducted? How long? Time period/ year/ duration of the study?

Response: The data of this study were collected from May 31 to June 6, 2022 for one week and online survey was posted to recruit study participants via K university website that can be accessed only by their students (undergraduates and graduates). This content was added in the revised manuscript.

2) In the abstract, authors have mentioned age 19-30 whereas in method section it says 18-30. Please correct it.

Response: Thank you for catching the typo. Age 19-30 is right. The error was corrected.

3) Is there a rationale of selecting the particular age group?

 Response: This research targeted young adults who are less likely compliant with preventive health behaviors than older adults in order to identify possible factors and the selected age group (19-30) is typically in higher education (college and university) in Korea.

4) Mention about how participants confidently was maintained?

Response: How to maintain participants’ sensitive information was added in red to the revised manuscript. Thank you.

5) In methods section, there is a mention of self-reported questionnaire. Is this adopted from somewhere? Or is it self-made? You can add questionnaire as appendix. 

Response: We used four instruments: Three of them were adopted from different scholars and one was created as mentioned in the manuscript. Please refer to the methods section of the revised version.

We appreciate all your support indeed.

Reviewer 3 Report

Thank you for the opportunity to review the paper. COVID-19 is still an important issue in research. I've got, however, some doubts regarding the methodology:

- the group is rather small, why the Authors have decided on 91 participants only? Even if, according to the sample power 80 participants would be enough, there are still a lot of factors that might be missed in this type of approach.
- there's a lack of information about the level of education, profession etc. which might influence attitudes toward COVID,
- there are differences between the abstract and Methodology sections regarding the age of participants, (what was the minimum age?)
- why ages 25 and 26 were the criteria for the different two groups in the analysis?

Author Response

Thank you for the opportunity to review the paper. COVID-19 is still an important issue in research. I've got, however, some doubts regarding the methodology:

1) The group is rather small, why the Authors have decided on 91 participants only? Even if, according to the sample power 80 participants would be enough, there are still a lot of factors that might be missed in this type of approach.

Response: We, researchers, tried to gather more participants, but all we had were only 100 participants for a limited time. Out of 100 participants, 9 participants’ data were removed due to missing response to certain questions.

2) There's a lack of information about the level of education, profession etc. which might influence attitudes toward COVID.

Response: All participants were either an undergraduate or a graduate student. Accordingly, they have at least high school diploma or baccalaureate’s degree. Therefore, researchers did not specify either their level of education or professions. This time researchers focused on four variables, but we can collect more information of participants’ demographic aspects with a similar design for the future. Thank you for your comment on this point.

3) There are differences between the abstract and Methodology sections regarding the age of participants, (what was the minimum age?)

Response: Thank you for catching the typo. Age 19-30 is right. The error was corrected.

4) why ages 25 and 26 were the criteria for the different two groups in the analysis?

Response: There is no specific reason for this, and we selected these ages because they are middle numbers of the group.

We appreciate all your support indeed.

Round 2

Reviewer 1 Report

The improvements introduced in the new version are appreciated.

However, most of the observations of the initial review have not been followed up or applied; with which, the initial recommendation to reject its publication is maintained.

Observation: in order to recommend its possible publication, the authors must respond to each and every one of the objections made in the first revision, therefore, the indicated objections must be checked one by one and applied in the new version.

Author Response

Comments and Suggestions for Authors

The improvements introduced in the new version are appreciated.

However, most of the observations of the initial review have not been followed up or applied; with which, the initial recommendation to reject its publication is maintained.

Observation: in order to recommend its possible publication, the authors must respond to each and every one of the objections made in the first revision, therefore, the indicated objections must be checked one by one and applied in the new version.

Response: We, authors, responded to your comments line by line as you requested when responding to your comments. We are wondering if you missed our response to your comments when you reviewed the revised version. We also attached the supplementary document to have you understand better. Please refer to below responses that were already made as well as the supplementary document attached here.

Response to Reviewers’ Comments

Reviewer’s Comments and Suggestions for Authors

It is a topic and a very interesting, current and theoretically and socially relevant work. However, the article needs to be improved in many ways before it can be recommended for publication:

1) The background must be updated with the most recent references and empirical evidence (2022) and remove old references. In addition, the background must conclude in the research question; specified in the objective and materialized in the hypotheses or expected forecasts.

Response: We, researchers, truly appreciate your comments. According to your feedback, references are updated. However, some old references are used to define or explain terminologies, which are very crucial studies and remained in the revised manuscript. Also, they are classical references, mainly original studies of development of instruments, which need to be cited in the manuscript. Also, this study aimed to explore the relationship between variables, hypothesis was not created. Yet, we wrote the study objective more clearly as we respect your advice.

2) The presentation of the methodology should be significantly improved. --- The participants must be clear, including in a nested way the number x gender x other characteristics; In addition, the inclusion and exclusion criteria must be made explicit; the representativeness of the sample, etc.

Response: Inclusion and exclusion criteria were added in the section of materials and methods. Detailed description of participants’ characteristics was also written in results.

3) The description of the instruments must be complete, providing a table with them and their factorial weights, for example. In addition, with the study data itself, coefficients on composite reliability (not only for internal consistency or Cronbach's alpha), MacDonald's omega, the average variance extracted (AVE or convergent validity) must be provided; discriminant validity (square root of the convergent and that its coefficients are greater than the intercorrelations between the factors). Evidence of construct validity should be provided with the data itself, and if possible, confirmatory factorial analysis.

Thank you for your comments. There are four tools used in the study:  assessment of Optimistic bias, Hypochondria, Mass psychology, and preventive health behavior. Three out of four (Optimistic bias, Hypochondria, & Mass psychology) were not developed in this study. They were developed before and have been used in many studies. Therefore, Cronbach's alpha for each tool was written.  For the tool to measure preventive health behavior, researchers made 6 question items after reviewing the COVID-19 related preventive measures of WHO. Content validity and construct validity were tested and accepted to use. Exploratory Factor analysis for Preventive Health Behavior was conducted, and two factors were shown. However, these were COVID-19 related preventive health behaviors, which were not labeled. Researchers thought this content was to be explained and added ‘This instrument was made based on the preventive measures on COVID-19 suggested by the WHO, which included 6 items. All items were evaluated for content validity by 3 nursing professors on a scale of 1 to 4 points. Content validity index (CVI) of all 6 items was 3 points or higher. Exploratory factor analysis conducted for construct validity revealed that all 6 items had communality of 0.5 or higher, with KMO (Kaiser-Meyer-Olkin) of .704, Bartlett χ2 of 124.072, df of 15, and p of .000, which showed the appropriateness of the instrument.’ in the revised manuscript. Although we did not add the table of Exploratory Factor analysis for Preventive Health Behavior in the manuscript, it is attached here for your information.

4) A section should be introduced prior to the results, on data analysis and software used, but with more explanatory arguments, greater detail, why these analyzes are used and not others, and what they contribute. In this sense, presenting data that "go together", pure correlations, does not allow any statement to be made about the causal relationship between the measures. It would be desirable for them to carry out a causal analysis: I suggest two possible alternatives; One could be through the Hayes PROCESS macro, which is inserted in the Regression Analysis module and uses bootstrapping and is open access and very easy to implement; this would make it possible to provide mediation, moderation, or conditional analyses. Another alternative, through AMOS (or another program), through SEM (structural equations); that would allow establishing a model of causal relationships that would be much more interesting and generalizable than what they provide in the study.

Researchers aimed to identify causal relationship by using regression analysis, but found out that planned to conduct this study, four main variables (preventive health behavior, optimistic bias, hypochondria, mass psychology) didn’t have normal distributions based on Kolmogorov-Smirnov and Shapiro-Wilk tests. Accordingly, parametric test was not applicable and non-parametric test was used. Mann-Whitney test was used instead of T-test, Kruskal-Wallis test instead of ANOVA, and Spearman’s rho test instead of Pearson correlation. Regress analysis and SEM you suggested can be used when data are normally distributed as one of assumptions, which was not applicable to the data of this study. However, we attached the result of various regression analyses for your information. If you can provide rationales that can override this fundamental assumption for regression analysis, we are willing to add the results from regression analyses. Since the study could not identify causal relationship due to the data characteristics, the title of the manuscript was changed to ‘Relationship of preventive health behavior, optimistic bias, hypochondria, and mass psychology among young adults in Korea’. We also restricted expression that implicated causal relationship in the manuscript. Please refer to the Data analysis section and limitation of study in discussion that addressed this issue.

5) The presentation of results should improve notably. The steps indicated and explained before in the Data Analysis section should be followed. Of course, table 1 (and several of the following) is a simple list; so they don't do much for explanatory purposes. You should consider presenting complete nested tables, including not only the mean or median scores, statistical significances, effect sizes (practical value of relationships); and above all, to completely reorganize it to answer the research question that should allow causal analyzes (SEM type, PROCESS type?).

There is no change in the analysis method for the above reason in (4). Please refer to the response to (4).

6) The Discussion and conclusions must be reorganized and rewritten. It must answer the research question or problem of the study; it must indicate if the objective was achieved; whether or not the hypotheses are confirmed. An interpretation must be made in line with recent references and studies (2022); the limitations and their way of resolving them must be assessed; applications for practice; and the added value of the study or conclusions.

References were updated and the entire manuscript was thoroughly reviewed and revised.

7) The abstract must be rewritten according to these changes, clearly indicating the sample (N), what was done, the results and the added value. They must assess whether to also adjust the title to the new changes in the article.

The abstract was reviewed, and the title was changed to be consistent with the findings of the study.

8) Review all the formal and editorial aspects, articulating the article much more in its various parts. For example, update to the last date you review the article (or the latest possible before its publication) the reference 1 on the statistics used.

Thank you for your feedback. The last date of reviewing the article and the access date will be updated right before its publication.

9) All changes in the new version must be marked in color, to facilitate their verification. In addition, to argue one by one in the magazine's forum for the reviewer; so that what has been done can be properly understood and valued.

The revised areas of the manuscript were written in red with track changes for you to easily review. Also, response was written under each comment of reviewers as seen.

Reviewer 3 Report

I've got only one more comment:

4) why ages 25 and 26 were the criteria for the different two groups in the analysis?

Response: There is no specific reason for this, and we selected these ages because they are middle numbers of the group.

It should be better explained in the Methodology section.

Author Response

Thank you so much for your feedback.

We, authors, added explanation about the age in the manuscript (result section) based on your comments.